# Is Pet Health Insurance Able to Improve Veterinary Care? Why Pet Health Insurance for Dogs and Cats Has Limits: An Ethical Consideration on Pet Health Insurance

**DOI:** 10.3390/ani12131728

**Published:** 2022-07-04

**Authors:** Michelle Becker, Holger Volk, Peter Kunzmann

**Affiliations:** 1Stiftung Tierärztliche Hochschule Hannover Institut für Tierhygiene, Tierschutz und Nutztierethologie (ITTN), Bischofsholer Damm 15, Gebäude 116, 30173 Hannover, Germany; peter.kunzmann@tiho-hannover.de; 2Stiftung Tierärztliche Hochschule Hannover Klinik für Kleintiere, Bünteweg 9, Gebäude 280, 30559 Hannover, Germany; holger.volk@tiho-hannover.de

**Keywords:** pet health insurance, animal ethics, animal welfare, price discussions

## Abstract

**Simple Summary:**

The standards of veterinary medicine are increasingly approaching those of human medicine. As a result, the cost of veterinary medicine is also rising noticeably. For many veterinarians and also pet owners, the question therefore arises as to whether animal health insurance is a possible solution. Based on a thought model, we have classified pet owners into four groups according to their willingness to pay and their dispensable funds. We found that animal health insurance can reduce price discussions, but reaches its limits as soon as an animal owner can afford neither the veterinary costs nor a monthly insurance premium.

**Abstract:**

Background: Owners often feel the cost of veterinary care is too high, as there remains a limited understanding of the cost of health care in human and veterinary medicine alike. Pet health insurance is often seen as a universal solution. However, especially for patient owners with few financial resources, both the bill at the vet and the monthly premium for pet health insurance can become a challenge. Hypothesis: Pet health insurance can prevent or ease many price discussions at the vet, but it does not offer a solution for patient owners with little financial means. Methods: In order to verify for which patient owners pet health insurance can be a solution, four theoretical groups were formed depending on the patient owner’s willingness to pay and his/her dispensable funds based on a theoretical model. Results: Dispensable funds are a factor that cannot be influenced by the veterinary surgeon. However, low dispensable funds as a result of an insufficient willingness to save (whether due to a lack of financial education or a lack of will) can be solved by pet health insurance. Willingness to pay, on the other hand, can be influenced by empathetic communication from the veterinary surgeon and thus also from pet health insurance. Nevertheless, situations remain where pet health insurance is not a solution either, because owners can neither afford the veterinary costs nor a premium for a pet health insurance.

## 1. Introduction

Financial discussions occur daily in small animal practice [1], which can be frustrating for the whole veterinary team, but for the veterinary surgeons in particular. Owners often feel the cost of veterinary care is too high, as there remains a limited understanding of the cost of health care in human and veterinary medicine alike.

Dealing with financially limited clients can quickly become a burden for veterinary surgeons and increase the stress level at work [2,3]. This is in part due to veterinary surgeons wanting to provide the best possible care for each individual animal, which clashes with owners’ financial constraints and the misconception of some owners that veterinary surgeons are mainly motivated by making a profit.

Kondrup’s results show that ethical issues arise, affecting veterinarians personally and guiding the selection of treatment options [1]. The veterinary profession has advocated for strengthening the pet health insurance market to improve pet health care for the benefit of all stakeholders [2,4].

Can pet health insurance really solve most of the challenges of small animal practice mentioned above? The stakeholders in a veterinary appointment are the animal, the owner and the vet. What are their needs, and is pet health insurance able to better fulfill those needs?

Veterinarians and owners have a common interest in the welfare of the animal.

According to the RCVS code of professional conduct [4,5], it is part of the veterinary profession to “ensure the health and welfare of animals”. This is also part of the declaration every veterinary surgeon makes. The owner can also be said to have a fundamental interest in the welfare of his animal, since in many cases the relationship with his animal can be described as similar to that of his own child. [6]

Clients’ expectations were also identified by Kirsty Hughes et al. [7]. They designed a “Veterinary client hierarchy of needs”, which defines clients’ needs as follows:The animal’s welfare;Clinical problem-solving ability;Professionalism;Communication skills;Working in partnership.

“Some capabilities are considered essential while others are considered as valuable add-ons once the fundamentals are in place. It may be that our clients coming to see the vet that their first priority for their animal is its safety and physiological needs and that once they see those being met, they can then prioritize their own psychological needs”.

In addition, both in the UK and in Germany, pet owners are legally obliged to care for their pets.

The UK legislation defines the duties of a “person responsible for an animal to ensure welfare” as follows [8]:

“(2) For the purposes of this Act, an animal’s needs shall be taken to include—
(a)its need for a suitable environment,(b)its need for a suitable diet,(c)its need to be able to exhibit normal behaviour patterns,(d)any need it has to be housed with, or apart from, other animals, and(e)its need to be protected from pain, suffering, injury and disease”

German legislation sees the protection of animals as a human responsibility and holds that “pain, suffering or harm” [9] to animals can only be applied with a legitimate and justified reason. Thus, from a legislative perspective, pet owners should be able to care for their pet adequately in health and disease. Legislation provides a basis for deciding how costs can be paid in an emergency before the emergency occurs. Otherwise, dilemmas can arise for all parties involved, which can put a heavy burden on the parties involved (animal, owner, veterinarian).

As mentioned in UK legislation, German legislation defines the responsibilities of pet ownership as follows: Your duties are to offer a diet, care and accommodation according to the needs of the pet. You may not restrict species-appropriate movement if this causes unnecessary suffering and pain. You have to acquire appropriate knowledge about the points mentioned above [9].

The animal’s expectations can be defined as the fulfilment of basic needs, such as the Five Freedoms, but also the quality and quantity of life should be maximized. The Five Freedoms are defined as freedom from hunger and thirst; freedom from discomfort; freedom from pain, injury and disease; freedom to behave normally; and freedom from fear and distress [10]. The freedom from pain, injury and disease can especially be addressed by pet health insurance.

Veterinary surgeons often find themselves in a difficult situation, on the one hand wanting to care for the animal, and on the other hand considering the owner’s financial capabilities and value system. It is a conflict between the original decision of becoming a vet, with fulfillment of the animal’s needs as the highest value, and letting owners decide which diagnostic and therapy they want to choose. “For in any veterinary consultation in any branch of the profession, there are three interested parties: the client, the animal and the practitioner him- or herself” [11]. One possible solution for this conflict may be understanding the patients’ needs. These are as described by Coe [12] as follows:

“*Care of the animal should take precedence over monetary aspects.*[…] There was an expectation among some participants that out of a shared interest for the pet, the veterinarian would work with the client to find a solution if the client could not immediately afford veterinary care. […]
*Discussions of costs should be initiated upfront.*

*Costs of veterinary care should be placed in a meaningful context.*
[…] Costs should be discussed within the context of their pet’s health and prognosis, stating, for instance, that “I want the information about cost in the context of what’s a reasonable prognosis.”
*Client suspicion should be addressed.*
[…] The most consistent suspicion arose from the conflict between the idea of veterinary medicine as a health-care profession versus a business.”

Meeting the needs of clients is important to satisfy both veterinary surgeons and clients in daily veterinary practice. It may also make a huge difference for a better working environment with less daily conflicts.

Johanna Kersebohm [4,13] has identified vets’ expectations of their workplace and working environment. The two most important ones are a good working atmosphere and an appropriate salary. These two factors are largely determined by how the vet experiences their time in consult: most of the time is spent in consults every day, which are also the main contributors to the revenue of a practice. During this time, discussions about cost and ethical dilemmas can arise [14].

Insurers, as one of the stakeholders of pet health insurance, see an interesting niche market in Germany that they would like to tap into. They are aware of the problem of unpaid veterinary bills and would like to establish veterinarians as multipliers in order to create a win-win situation [15].

Despite pet health insurance being a solution to improve pet health care, many owners choose not to take out a policy for their pet due to financial constraints or restricted coverage. The current study investigates if a voluntary pet insurance system could resolve these conflicts and provide accessible care for pets. Would this have a direct effect on animal welfare (“According to studies, dog owners with pet health insurance spend 29 per cent more annually for veterinary care; cat owners 81 per cent more.” [16]) or only an indirect one by enabling the vet to have discussions with the owners which are not financially restricted and to see cases on regular routine health check-ups?

As mentioned above, veterinary surgeons have a fundamental interest in the wellbeing of their patients. However, in contrast to the patients’ owners, there is one additional point: to be able to work in the profession in a financially sustainable and healthy way (physically and mentally).

The aim of this work is to find out if pet health insurance is really as helpful as is claimed in veterinary press.

“Since the 2005 change in the law preventing veterinary surgeons from recommending particular insurance products, we have lost a very useful symbiotic relationship with the insurance industry”(GB) [17].

“[…] [Owners] will no longer have to worry whether or not they can afford the necessary veterinary attention. If an animal is insured by a caring owner for this laudable reason, it frees us to consider only the animal’s health and welfare and ensure we reach an accurate diagnosis and satisfactory conclusion […]”(GB) [4].

## 2. Methods

To find out how to solve price discussions, it is worth taking a look at their origin. This work is based on a theoretical model.

Due to the fact that surveys often distort the picture of reality, we have deliberately chosen a thought model. This effect has already been demonstrated in the area of price sensitivity for fresh meat. In surveys, the majority of consumers stated that they would accept a price premium of just over 30% if more animal welfare could be guaranteed [18]. In a supermarket study, however, this willingness to buy was disproved [19]:

“The results are surprising, as various surveys have shown that many consumers are willing to spend significantly more money on meat if it has been produced according to higher animal welfare standards. The results of the present suggest that the observed reality of actual purchasing behavior is more differentiated and complex. The basic willingness to spend more money on such meat in the test for such meat is only pronounced to a limited extent. General statements on willingness to buy should therefore be viewed critically.”

Although the consumption of meat is a voluntary purchase, while consulting a veterinarian in the event of illness is not a voluntary decision, the parallel is intended to illustrate how large the gap often is between a statement made and actual action in society.

Therefore, the added value of this theoretical model is that, supported by the existing literature and a theoretical analysis of the current situation, potentials and limitations of animal health insurance can be shown.

The basic idea of the following thought model is that price discussions often arise for two reasons: either the owner does not have the financial means to pay, or he does not want to use the required resources.

Thus, financial conflicts can be reduced to two fundamental factors: willingness to pay and dispensable funds.

These two factors combined lead us to four different situations. Thus, we have developed a model which helps to classify patient owners in terms of their willingness to pay in relation to their financial capacity. Based on this model, predictions of pet health insurance limitations and capabilities can be drawn.

The two factors “willingness to pay” and “dispensable funds” of patient owners can be combined into four different categories:High willingness to pay (WP) and high dispensable funds (DF);High WP and low DF;Low WP and high DF;Low WP and low DF.

WP depends on the human–animal bond. When the relationship of the owner with the animal is very close, the owner is more willing to spend money on her or his pet. (According to the NAPHIA Press Kit, pet owners spend more at the vet when they have pet health insurance in place. Reasons for purchasing PHI include that it “is helpful to pet owners,” “shows you love your pet” or “helps avoid the need to make painful choices about care.” In other words, many reasons for taking out pet health insurance indicate a close relationship with the pet and are intended to open the door to expensive diagnostic and treatment methods [16].) In the media you often find only positive statements about the relationship, especially between man and dog [20,21], but this relationship can be very complex [22]. Owners rarely would admit that they have a more practical, less caring relationship with their pet.

## 3. Results

Four groups of patient owners could be identified in the model above.

Group 1: high WP/high DF

This owner loves his pet and is willing to pay all the bills that come with it. This group is found in many households that acquire a dog or cat which lives with them inside the house. Health insurance for the pet is self-evident, as the vet has recommended it and the owners want only the best for their pet.

Group 2: high WP/low DF

This owner also loves his animal, but has limited dispensable funds available. Often the animal is an emotional support or at the time of purchase the awareness of possible costs and the responsibility to care for the animal was low. This group is often found in socially weak households. In this case, if the owner is aware of possible veterinary costs, PHI is a good way to cushion financial peaks and convert them into monthly amounts.

Groups 3 and 4: low WP/high DF and low WP/high DF

This owner has a low emotional attachment to his pet. This may be the case if the animal has a specific task. In the case of dogs, for example, this can be the herding of sheep or personal protection. In the case of cats, this could be keeping away pests such as mice or rats. As part of the pet owner’s responsibility for the animal, a monthly payment might actually be more convenient for these owners than individual vet bills, as it makes costs more predictable.

A low willingness to pay can also occur if not all family members agreed with the acquisition of the animal.

Groups 3 and 4 differ only in the (non-)presence of financial means.

## 4. Conclusions

As shown in Figure 1, pet health insurance is not needed for the high WP/high DF group, but is a solution for owners who have been categorized either in high WP/low DF, low WP/high DF or low WP/low DF. However, for the high WP/low DF or low WP/low DF group the question remains if pet ownership is in general justified.

In the high WP/low DF group, owners would be willing to pay but their means are restricted, leading to the high frustration of owners and vets alike. The low DF is due to the fact that the income or assets are generally too low to pay for veterinary bills.

In Germany, anyone who “keeps, cares for or has to care for an animal” is legally responsible for “feeding, caring for and housing the animal in a manner appropriate to its species and needs” and for acquiring the relevant knowledge. This includes nutrition, care (also in case of illness) and accommodation, but also the necessary “knowledge and skills” [9]. In the Animal Welfare Act, it is equally written that owners must protect their animal from pain and suffering [8]. It is therefore questionable in this case whether keeping animals is ethically justifiable.

Another scenario for a low DF is a low willingness to save money. Most owners are aware of the annual costs for a general health check, vaccinations, flea/tick treatment and deworming when they purchase a pet. However, there is a limited understanding of costs which arise due to an illness. Most people have neither the foresight nor knowledge of how high veterinary costs can be. If that were the case, there would be no price discussions and fewer people would keep an animal unless they consciously accept the risk of a veterinary shortage. They might also not have the discipline to set aside a fixed monthly amount. Even if an owner does save money health costs for their pet, it might exhaust their savings quickly. It is therefore obvious to consider animal health insurance as a possible solution.

The relationship of the low WP/high DF group to the animal is rather pragmatic. It is possible to keep the animal for a certain benefit, for example, a cat to keep the mice at bay or a guard dog for personal or property protection. Veterinary expenses are limited in this case to keeping the animal healthy, always with a cost–benefit calculation being carried out, which of course quickly reaches its limits when higher expenses are incurred. It is important to inform the owner that keeping an animal is also associated with the obligation to care for it in a species-appropriate manner. This naturally includes adequate veterinary care. Paragraph 1 of the Animal Welfare Act provides the basis here: “The purpose of this Act is to protect the life and well-being of the animal as a fellow creature out of the responsibility of man for the animal. No one may inflict pain, suffering or harm on an animal without reasonable cause” [9]. As mentioned above, the UK’s Animal Welfare Act demands freedom from “injury and disease” [8].

Predictable costs are easier to accept than one-off, large sums of money that are covered in an unscheduled manner and appear to be lower overall than the full amount of costs. This is the reason why instalment payments are very popular with consumers [23]. Pet health insurance, together with legal and ethical argumentation, can therefore certainly represent a solution here as well.

Group four has both a low WP and a low DF, so if there are no available funds because the money has been spent on something else, pet health insurance could be a solution in that the patient owner can be convinced of the relevance of the product and monthly costs are better accepted than isolated monetary spikes.

If there are no available funds due to low income and a lack of reserves, the question arises as to why the owner owns this animal. If the farm is supported by dogs and cats (herding dogs, guard dogs), then in the absence of financial resources, the starting point is not likely to be PHI.

In summary, the following possible scenarios can be identified here which could potentially be solved by pet health insurance.

An insufficient willingness to save (whether due to a lack of financial education or a lack of will) can be solved by pet health insurance. The monthly contribution is then deducted from the account without the owner’s intervention. A low WP in combination with a low DF could also be solved by pet health insurance. In all situations the decision for or against pet health insurance or the importance to inform about it remains the same. The fact that the veterinarian plays a major role in the advice given has been demonstrated, at least in North America: “The survey among pet owners demonstrated that 50 per cent more pet owners would likely purchase pet health insurance if their veterinarians actively recommended it” [16].

Pet health insurance could bring added value to animal welfare in the mentioned situations. Animal welfare can be improved if the choice of the optimum diagnostic and therapeutic method can always be made on the basis of ethical considerations rather than financial limitations, and if an animal is regularly presented for a preventive health check-up [24]. Some pet owners do not bring their pet for a regular health check-up due to financial concerns: “In a study of pet owner expenditure, Henderson […] found that financial issues were a barrier for pet owners when it came to preventive, sick, and emergency care” [25]. However, early detection of diseases can be essential and significantly improve the quality of life.

Still, situations remain where pet health insurance is not a solution either, because owners can neither afford the veterinary costs nor a premium for pet health insurance.

As described in depth here, pet insurance could help to cover the costs of veterinary care. However, the actual costs might not be the biggest challenge for the veterinary profession. According to Brennecke and Münow, the reason for a lack of acceptance of prices is not to be found in the price itself but in the way a patient owner feels treated. Guido Bentlage writes about this: “Above all, owners want to be treated friendly. They want a committed, caring veterinarian who takes time for them and their animal. […] Contrary to what most veterinarians actually expected, the price, i.e., the costs, was even at the bottom of the ranking. As already mentioned, friendliness was by far in first place” [26].

“The focus or, rather, the entire objective of veterinary activity should be on customer satisfaction, in which medical quality, although a very important part, is just one part.”

“Pet owners’ expectations according to ranking (Brennecke 2009)” [26]:Friendliness;Attention;Helpfulness;Telephone/personal availability;Speed;Reliability;Professional/social competence.

As we have seen in the results of Kirsty Hughes, it is important to first build a base: the animal’s welfare plays a fundamental role. Only if this need of the owner is fulfilled is there the opportunity to win over the owners with friendliness and empathic communication and to profit from the interaction due to professional competence and customer-orientated service.

Information about the costs involved is also part of good advice and even contributes to customer satisfaction [26]. Empathic communication and active listening can reduce a large part of both price discussions and the resulting ethical dilemmas, as well as having a positive effect on the economic profit of a veterinary practice [27].

Vets often focus too little on costs and see costs as a limitation to better pet health care. Pet health insurance seem to be an easy solution to address discussions around costs. However, as aforementioned, owner needs are not cost-related, but perceived value-related, and can be addressed with professionalism and verbal and non-verbal communication with empathy, helping to build a trustful relationship between the veterinarian and patient owner [28]. Therefore, the pet health insurance debate might be over-inflated and play a smaller role than anticipated. Moreover, unless pet health insurance would be obligatory, mainly owners with appropriate financial means will spend their disposable funds for health insurance, and they are most likely also able to afford care without insurance.

The situations that really pose a challenge are those where the patient owner really has no money for treatment. As filtered out in the model, some of these patient owners can make the money available through predictable payments for pet health insurance, but those who cannot do so cannot afford either treatment or pet health insurance and are thus excluded from the scope of pet health insurance. The ethical question, however, remains if these individuals should have a pet if their funds are so limited. Dispensable funds are a factor that cannot be influenced by the veterinary surgeon.

Willingness to pay, on the other hand, may be influenced by empathetic communication from the veterinary surgeon. Communication needs to be improved not only in the consult room, but also to the public, so there would be a better appreciation and understanding of the cost of pet health care.

### Further Perspectives and Conclusions for Practice

As described in the article, PHI can be a positive contribution in the field of small animal medicine and serve as a valuable support in the treatment situation of small animal medicine. However, in Germany, so far significantly fewer dogs and cats are insured than is the case, e.g., in UK.

So how can the product be made more attractive so that more pet owners take advantage of it?

Looking at the general functioning of an insurance company, it is a characteristic of the system that with a larger pool of insured persons, the monthly premiums decrease due to a risk equalization: “The [risk equalization] in the collective finds expression in the fact that, under otherwise constant conditions, a growing collective size offers advantages. These advantages consist in the fact that, as the size of the collective grows, either the random risk itself is reduced or, with a constant (controlled) level of security, the insured total loss can be financed more favorably on average, making it cheaper for the individual policyholder to purchase insurance coverage” [29].

This means if the product were more attractive (both in terms of price and content), more people would likely take advantage of it. However, this only works with an increased pool of insured dogs and cats.

Would mandatory health insurance for animals be a possible solution at this point?

Fundamental challenges are already encountered in the design of the content [30]. Considering the models of health insurance from human medicine as a basis, a mixed model of Bismarck and Beveridge would be conceivable. PHI would therefore be subsidized from government funds. This state of affairs can be questioned as quite unrealistic, at least in the near future for our pets. Thus, excluding this subsidy, we are left with a pure Bismarck model, i.e., payment of health care costs from premiums without government subsidy. While this fact in itself sounds superficially feasible, it does not solve the fundamental challenges mentioned at the outset, which essentially relate to the design of exclusions and the assessment of premiums.

How can the findings now be profitably used in practice?

As the NAPHIA Press Kit recommends [16], the veterinary practice is the number one source of information for animal health insurance. With education about potential veterinary costs and recommendation to inquire about PHI, more pet owners will be made aware of the benefits of pet health insurance.

## Figures and Tables

**Figure 1 animals-12-01728-f001:**
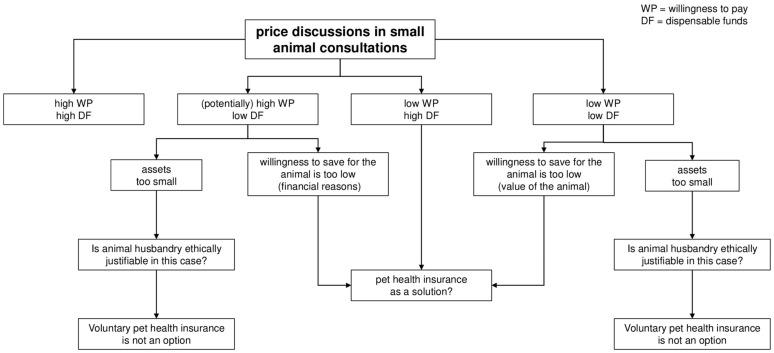
Patient owners’ willingness to pay depending on dispensable funds.

## Data Availability

Not applicable.

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
