# Peer review of "Is Pet Health Insurance Able to Improve Veterinary Care? Why Pet Health Insurance for Dogs and Cats Has Limits: An Ethical Consideration on Pet Health Insurance"

_animals, 2022, doi:10.3390/ani12131728_

Round 1

Reviewer 1 Report

Thank you for the opportunity to review this manuscript. I commend the authors on the importance of the study topic. I think this manuscript provides novel data that can improve the understanding and role of pet health insurance. However, I have some recommendations for the authors regarding the framing and examination of health insurance, itself. I recommend this manuscript for major revision, given the importance of the topic and the novelty of the data. Yet, the current framing does not provide an accurate understanding of the mechanisms of how health insurance works. Revising this in the introduction and conclusion would significantly improve the manuscript. 

Major Comments:

1. As it currently reads, the article shows a limited understanding of the principles of health insurance. Effective health insurance relies on sufficient payer pools (a large number of clients, here, pets) in order to reduce the cost risks to the entire pool (distribute the cost burden). For example, consumers pay 60 dollars a month to have 90% of medical costs covered, pay 10% out of pocket. Lower risk clients (e.g., younger, fewer or no pre-existing conditions), pay less. Higher risk clients (older, sicker) pay more. In human health policy, governments use subsidies and policies to make insurance more equitable, to ensure that sicker and poorer populations aren't just shouldering a high cost burden that would make care inaccessible. This comes in the form of subsidies and regulations to offset monthly costs (premiums) and out of pocket costs (the part that insurance doesn't pay when you get a medical bill). The LARGER the insurance pool, the greater amount of money available to draw from to cover the costs of the sick individuals when they get sick, and thus, the lower the monthly costs to the individuals who are paying into the monthly insurance payments. Thus, any and all insurance models rely and are conditional on who is in the pool (how many, and what their risk level is). The real question here that this article must address is whether or not such conditions can be created for animals as they have been for humans. If not, animal health insurance will remain expensive and out of reach for most people. 

2. Per the example in UK/Germany on pg. 2: the legislation and court rulings cited place responsibility on individuals, rather than on systems or societies to provide health care for animals. This inherently means that any insurance model will be insufficient, unless their is a way to incentivize sustainable risk pool buy in from pet-owners. This logic must change in order for the models to work. Otherwise, they will always only work for a small subset of the population that is willing to pay higher out of pocket costs (necessitated by a small risk pool). 

3. This study could be useful if the authors acknowledge that they are looking at a voluntary pet health insurance system, and if the authors are very clear that this voluntary system would be absent any incentives (to bring more people into the pool) or subsidies (to offset costs for people who can't afford monthly premiums) and they are looking solely at owner willingness to pay. This also might offer insight into the likelihood that future models of voluntary pet health insurance might become more sustainable, if there is enough buy-in initially, from enough owners (with younger, healthy pets and fewer medical costs, to lower the price of premiums over time). 

4. pg. 4, line 149 "this effect has already been demonstrated in the area of price sensitivity for fresh meat" - I highly suggest the authors review the extensive literature on human health insurance. Meat is not relevant. This is not an example of health insurance, but an example of a market commodity, which health insurance is explicitly designed to protect against. Health care doesn't work like other market goods because it is inelastic - you get care when you need it, if you go without it or if animals go without it, then they die or get sick. Meat you can choose to purchase or not, or go to a place that is less expensive. You can't do this in a medical emergency for people or animals. Thus, the theoretical assumptions of the model are incorrect. The history of health insurance with merchant mariners https://www.jstor.org/stable/44446303 is a good place to start to understand how health insurance works.

5. pg. 5 Figure 1: This figure is confusing because the introduction frames the paper as a mandatory insurance system. This model does work for voluntary systems -- but the authors should be very clear in the introduction, methods and conclusion that this is what they are evaluating -- and acknowledge that voluntary systems may not work well because of inability to control risk pools (and thus may remain very high cost because of low enrollment and sicker animals with higher costs e.g., people enrolling their pets only when they have a medical emergency = very high premiums, less coverage for medical costs). 

Author Response

Thank you very much for the valuable revision of my paper. I have tried to implement all comments in the best possible way and I think the paper has gained a lot of added value from your thoughts.                

1. The real question here that this article must address is whether or not such conditions can be created for animals as they have been for humans

I added an additional section to explain how insurance works and how we can make it more affordable for more pet owners.
I have touched on the fact, that IF phi would have a positive impact on small animal medicine, why should it then not be mandatory.
For further thoughts on the mandatory phi, I wrote another paper (DOI:10.2376/0032-681X-2208).

Why I have inserted this section only at the end is based on the following consideration:
The main topic of the paper is to find out if phi can have a positive effect on small animal medicine in general. We then found out that there are some situations that are positively influenced by phi, but the premium itself can be a challenge for some owners. This is then followed by consideration of how these premiums can be made more favorable: Namely, by increasing the pool of insured animals and better spreading the risk.

2. The legislation and court rulings cited place responsibility on individuals, rather than on systems or societies to provide health care for animals.

It is correct that the responsibility lies with the owner.
It seems to be, that some owners don’t know, what veterinary costs you have to expect and which responsibility you own with an animal. This realization must be made, in order to decide whether to buy a phi or not. This article only asks, in which cases phi could have a positive impact.
If you think further and assume, that phi has a positive Impact for small animal medicine, then you can consider to make it mandatory. Until that point, the responsibility stays with the owner.
I made a supplement on page 3.

3. This study could be useful if the authors acknowledge that they are looking at a voluntarypet health insurance system

I think that was an oversight on my part. I have now corrected all passages that contained a mandatory instead of a voluntary phi, because this article deals with the voluntary phi.

4. Meat is not relevant

I chose the parallel with flesh to show the gap between a statement and an actual act in general. You are right when you say that buying meat is a voluntary decision, but the issue of veterinary costs should be decided before it becomes an involuntary decision.

5. This figure is confusing because the introduction frames the paper as a mandatoryinsurance system. This model does work for voluntary systems.

See point 3.

Reviewer 2 Report

Thank you for the opportunity to review this work. This work is definitely original and provides a theory that may potentially assist veterinarians in practice. I don't however think it fits the experimental design and scientific report format. I'd suggest you resubmit this work as a white paper proposing you theory/model based on the factors of WP and DF. I'd also like to see some additional data and/or description on how this theory/model was developed and how it might be used or applied. Consider providing more details on the four categories in relation to pet insurance. For instance a person with low WP, but low DF who acquires a healthy puppy and immediately signs them up for insurance might be able to maintain the subscription depending on the premium. Is this a third factor to be included in your model? How should the reader use this information? How does it really shape veterinary medicine? Finally it's important to mention that there are low cost clinics for low income and homeless persons in the US which I am sure are present  in other countries as well. With that said, I would rephrase your conclusion in lines 324 and 325 around the ethical question of pet ownership in those who can't afford care as this blanket statement is inaccurate if a person can pursue other avenues to receive the care their pet needs. Overall, I think this is a great paper that with some reshaping will contribute to the profession. 

Author Response

Thank you for reading and revising my paper. In response to your comments, I have tried to make a better reference to practice and add some further explanations.

I'd suggest you resubmit this work as a white paper proposing you theory/model based on the factors of WP and DF.

Do you think that this is covered by the simple summary?

I'd also like to see some additional data and/or description on how this theory/model was developed and how it might be used or applied. 

I added a final section to my paper, which tries to explain how to use this in practice: Phi might have a positive impact for most owners, most animals and veterinarians. So now the task of the profession is to inform about the topic and make more animal owners aware of the phi.

Consider providing more details on the four categories in relation to pet insurance.

I have tried to make a better reference to phi without prejudging the results.

For instance, a person with low WP, but low DF who acquires a healthy puppy and immediately signs them up for insurance might be able to maintain the subscription depending on the premium.

In this case, the first and most important step, is that the owner recognizes the necessity of a pet health insurance in his case. If this happens, the further treatment depends on regular precautionary examinations and the design of phi exclusions.

I have touched on this case on page 7 lines 296-304.

Finally it's important to mention that there are low cost clinics for low income and homeless persons in the US which I am sure are present  in other countries as well. With that said, I would rephrase your conclusion in lines 324 and 325 around the ethical question of pet ownership in those who can't afford care as this blanket statement is inaccurate if a person can pursue other avenues to receive the care their pet needs. 

From a patient owner responsibility perspective, it may be true that clinics providing free veterinary care for those in need are a solution. From the point of view of the veterinary medicine industry, this derivation seems to me not quite thought through to the end, as the financing of these clinics must be secured either by the veterinarians themselves (voluntary work) or by the state (tax money) or by donations. All three financing possibilities make the needy animal owners dependent on others. In addition, I submit that this care is a very basal one and once more extensive surgical care becomes necessary, it is limited (also for cost reasons). At least this is the case in Germany.

Round 2

Reviewer 2 Report

Thank you for taking the time to review and respond to my questions. I also appreciated the addition to the conclusion of the paper. I am excited to see how this article shapes the conversation around insurance and pet ownership given recent cases here in the US that have brought the topic of cost of veterinary care back to the forefront (https://wgme.com/news/i-team/mvmc-responds-to-i-team-investigation-maine-veterinary-medical-center-jaxx-german-shepherd). A true solve is still needed to protect the profession from undue criticism. Hopefully this work will at least help start the conversation. 

Author Response

Thank you for responding to my comments. 
I also hope that my work can provide inspiration. I am firmly convinced that pet health insurance will make the work of many veterinarians easier and that it will enable many animals to receive better veterinary care.